# Epigenetic Changes in Equine Embryos after Short-Term Storage at Different Temperatures

**DOI:** 10.3390/ani11051325

**Published:** 2021-05-06

**Authors:** Gustavo D. A. Gastal, Dragos Scarlet, Maria Melchert, Reinhard Ertl, Christine Aurich

**Affiliations:** 1Center for Artificial Insemination and Embryo Transfer, Department for Small Animals and Horses, University of Veterinary Medicine Vienna, 1210 Vienna, Austria; Maria.Melchert@vetmeduni.ac.at; 2Instituto Nacional de Investigación Agropecuaria INIA, Estación Experimental La Estanzuela, Ruta 50 km 11, 39173 Colonia, Uruguay; 3Division of Obstetrics, Gynecology and Andrology, Department for Small Animals and Horses, University of Veterinary Medicine Vienna, 1210 Vienna, Austria; dragos.scarlet@uzh.ch; 4Vetcore Facility, University of Veterinary Medicine Vienna, 1210 Vienna, Austria; Reinhard.Ertl@vetmeduni.ac.at

**Keywords:** equine, embryo, transport, methylation, development, embryo-maternal recognition

## Abstract

**Simple Summary:**

In embryos subjected to assisted reproductive techniques, epigenetic modifications may occur that can influence embryonic development and establishment of pregnancy. In horses, the storage temperature during transport of fresh embryos before transfer is a major concern. The aim of this study was, therefore, to determine the effects of two storage temperatures (5 °C and 20 °C) on equine embryos, collected at day seven after ovulation and stored for 24 h, concerning morphological development, expression of candidate genes associated with embryo growth and development, maternal recognition of pregnancy, methylation, apoptosis and gene-specific and global DNA methylation. Temperature during storage did not affect embryo size. There were no changes in pH and lipid peroxidation of the medium irrespective of group. mRNA expression and gene-specific DNA methylation of genes related to growth and development, maternal recognition of pregnancy, DNA methylation and apoptosis in stored embryos (5 °C and 20 °C) were altered when compared to fresh embryos. Therefore, our study demonstrates for the first time the gene-specific and global DNA methylation status of fresh equine embryos collected on days seven and eight after ovulation. Short-term storage, regardless of temperature, may compromise embryo development after transfer.

**Abstract:**

In embryos subjected to assisted reproductive techniques, epigenetic modifications may occur that can influence embryonic development and the establishment of pregnancy. In horses, the storage temperature during transport of fresh embryos before transfer is a major concern. The aim of this study was, therefore, to determine the effects of two storage temperatures (5 °C and 20 °C) on equine embryos, collected at day seven after ovulation and stored for 24 h, on: (i) morphological development; (ii) expression of candidate genes associated with embryo growth and development, maternal recognition of pregnancy, methylation and apoptosis, and (iii) gene-specific and global DNA methylation. Embryos (*n* = 80) were collected on day seven or day eight after ovulation and assigned to four groups: day seven control (E7F, fresh); day seven, stored for 24 h at 5 °C (E5C); day seven, stored for 24 h at 20 °C (E20C) and day eight control (E8F, fresh 24h time control). The embryos and the storage medium (EquiHold, holding medium, Minitube, Tiefenbach, Germany) from all treatment groups were analyzed for (i) medium temperature, pH, and lipid peroxidation (malondialdehyde; MDA) and (ii) embryo morphology, mRNA expression and DNA methylation (immunohistochemistry and gene-specific DNA methylation). The size of embryos stored at 5 °C was larger (*p* < 0.01), whereas embryos stored at 20 °C were smaller (*p* < 0.05) after 24 h. There were no changes in pH and MDA accumulation irrespective of the group. The mRNA expression of specific genes related to growth and development (*POU5F1, SOX2, NANOG*), maternal recognition of pregnancy (*CYP19A1, PTGES2*), DNA methylation (*DNMT1, DNMT3A*, *DNMT3B*) and apoptosis (*BAX*) in the E5C and E20C were either up or downregulated (*p* < 0.05) when compared to controls (E7F and E8F). The immune expression of 5mC and 5hmC was similar among treatment groups. Percentage of methylation in the CpG islands was lower in the specific genes *ESR1*, *NANOG* and *DNMT1* (*p* < 0.001) in E20C embryos when compared to E8F (advanced embryo stage). Therefore, our study demonstrates for the first time the gene-specific and global DNA methylation status of fresh equine embryos collected on days seven and eight after ovulation. Although our results suggest some beneficial effects of storage at 20 °C in comparison to 5 °C, the short-term storage, regardless of temperature, modified gene expression and methylation of genes involved in embryo development and may compromise embryo viability and development after transfer.

## 1. Introduction

Assisted reproduction techniques (ART) in horses have considerably advanced during the last decade. In this field, artificial insemination along with embryo transfer are still the most common ART applied to horses [1]. The production of embryos by ovum pick-up and subsequent intracytoplasmic sperm injection, however, is attracting increasing attention. Because of a limited availability of recipient mares at facilities that collect fresh embryos, or produce in vitro embryos, the shipment of such embryos is required with increasing frequency. Similarly, techniques that require specific laboratory equipment, such as cryopreservation, or require preimplantation genetic diagnosis, often demand overnight transport and sometimes transport back thereafter [2,3]. Equine embryos are transferred at day seven or eight after ovulation [1,4], with the majority developed to the blastocyst stage at the time of transfer [5]. For overnight transport, embryos are either cooled to 5 °C or kept at room temperature [6]. Therefore, transportation is a critical issue that can contribute to success or failure of subsequent embryo transfer.

During the development of preimplantation embryos, a wave of epigenetic reprogramming takes place to establish the totipotent state [7]. Epigenetics refer to certain hereditary DNA or chromosome alterations, such as methylation, histone modification and genome imprinting [8,9]. The reprogramming of DNA methylation is a necessary step for subsequent embryo development [10]. Processes involved in ART have been associated with epigenetic reprogramming disturbances [11], increasing the incidence of pregnancy complications [12] and imprinting disorders such as Beckwith-Wiedemann Syndrome (BWS), Angelman syndrome and retinoblastoma in children conceived with the help of ART [13]. In cattle, large offspring syndrome was described in calves derived from ART, which has been associated with loss of methylation and downregulation of certain maternally-expressed genes [14]. Embryo exposure to environments other than the uterus can alter cell metabolism [15] due to different types of environmental stress (e.g., temperature or exposure to chemicals; [16]), and the related oxidative stress can contribute to abnormal DNA methylation leading to impairment of embryonic development and failure in maternal recognition of pregnancy [16,17,18].

During early embryonic development, an increased expression of growth factors leads to proliferation of trophectoderm cells and prostaglandin production, likely involved in maternal recognition of pregnancy [19]. At the same time, DNA methyltransferases (DNMTs) play an important role in maintaining genome stability and integrity during development and epigenetic reprogramming [20], with *DNMT1*, *DNMT3A*, *DNMT3B*, and *DNMT3L* being the main genes involved in the establishment of methylation patterns required for cell lineage determination [21,22,23]. There is evidence that DNA methylation occurs mainly at the cytosines of cytosine-guanine dinucleotides (CpG) known as CpG sites [20]. Interactions between DNA and gene regulatory proteins can be critically influenced by the dynamics of DNA methylation. Genomic 5-methyl-2′-deoxycytidine (5mC) is recognized by methyl-DNA binding proteins that recruit histone deacetylases and can be reduced via oxidation to 5-hydroxymethyl-2′-deoxycytidine (5hmC) [24]. Therefore, changes in 5mC and 5hmC reflect global DNA methylation and hydroxymethylation in tissues and cells.

Whereas extensive information is available on epigenetic changes in oocytes and conceptuses derived from ART in humans, mice and cattle [24,25,26,27], information with respect to horses is scarce [28,29]. Our study followed the hypothesis that storage temperature contributes to epigenetic changes in equine embryos. To test our hypothesis, we evaluated if storage temperature of equine embryos alters the epigenetic reprogramming causing effects on gene expression of candidate genes associated with maternal recognition of pregnancy, embryo development, methylation and apoptosis involved in genetic reprogramming. Embryos collected on day seven after ovulation were either stored at 5 °C or 20 °C for 24 h. Fresh embryos collected on day seven or eight served as untreated controls. The following parameters were determined: (i) morphological development; (ii) gene expression of candidate genes associated with maternal recognition of pregnancy, embryo development, methylation and apoptosis, and (iii) gene-specific and global DNA methylation.

## 2. Materials and Methods

### 2.1. Animals and Reproductive Management

All experimental procedures were performed according to Austrian animal welfare legislation and approved by the Austrian Federal Ministry for Science and Research (license number BMWFW-68.205/0135-WF/V/3b/2014). Fifteen healthy and fertile Haflinger mares (4–16 years old) were used as embryo donors. Animals were kept in a large paddock with access to a shed, fed with hay and mineral supplements twice daily, and water was available ad libitum. The ovaries and uterus of mares were scanned transrectally with an ultrasound machine (Mindray M9, Mindray, Shenzhen, China) equipped with 5–8 MHz linear-array transducer (6LE5Vs) to detect a 3.5 cm preovulatory follicle and uterine oedema characterizing estrous, and to determine the time for artificial insemination [30]. Insemination was performed with extended (Equi Pro, Minitube, Tiefenbach, Germany) semen. One insemination dose contained at least 500 million progressively motile spermatozoa. Semen was collected from fertile stallions by artificial vagina using routine procedures [31]. The mares were inseminated at intervals of 48 h and checked every 24 h until ovulation was detected. No hormonal treatments were administered during the experimental period.

### 2.2. Experimental Design

Equine embryos (*n* = 80) were collected on day seven (*n* = 60) or day eight (*n* = 20) after ovulation and assigned to four groups with 20 embryos per group: (i) day seven control (E7F, fresh); (ii) day seven, 24 h at 5 °C (E5C); (iii) day seven, 24 h at 20 °C (E20C); (iv) day eight control (E8F). For short-term storage, embryos were kept in holding medium (EquiHold, Minitube) within an Equitainer (Hamilton Biovet, Ipswich, MA, USA), with or without freezer cans (E5C and E20C, respectively). The embryos and medium from all treatments were submitted to the following assessments: temperature, pH, lipid peroxidation, embryo morphology, mRNA expression and DNA methylation (immunohistochemistry and gene-specific DNA methylation).

### 2.3. Collection and Evaluation of Embryos

Embryos were recovered on days seven and eight after ovulation (day 0, ovulation detection) using an intrauterine silicone two-way Foley catheter CH 28 for mares (Minitube). The uterus was flushed four times with 1 L of Ringer’s lactate solution (Fresenius Kabi, Graz, Austria) prewarmed at 38 °C. The fluid recovered from the uterus was filtered through an embryo filter system (75 μm, EmCon embryo filter; Immunosystems, Spring Valley, WI, USA). The solution remaining in the filter cup was placed in a petri dish and analyzed under a stereomicroscope at 40 × magnification. Embryos were washed 10 times in holding medium (Minitube) to remove cellular debris. Embryos were measured before and immediately after the storage period under stereo microscope with an eyepiece micrometer, and morphologically classified according to their stage of development and quality on a scale from 1 (excellent) to 4 (degenerated) as described [32].

### 2.4. Holding Medium Temperature and pH

The temperature of the holding medium was recorded with a data logger (testo 175, Testo, West Chester, PA, USA) every 10 min for 24 h. Briefly, a second tube with the same volume of holding medium as for the embryo was placed inside the Equitainer, and the fine sensor of the data logger was placed inside this holding medium for temperature monitoring. The pH was assessed in a sample of the holding medium immediately before and after embryo storage for 24h. The pH was assessed by a pH meter (SevenCompact S220-micro-kit, Mettler Toledo, Columbus, OH, USA) using the microelectrode (InLab Ultra-Micro-ISM, Mettler Toledo, Columbus, OH, USA) for small sample volumes.

### 2.5. Lipid Peroxidation

Lipid peroxidation was determined by the reaction of malondialdehyde (MDA) with thiobarbituric acid (TBA) to form a fluorometric (λex = 532/λem = 553 nm) product, proportional to the MDA present, using a commercial kit (Cat#MAK085, Sigma-Aldrich Co., St. Louis, MO, USA) and following the principles and methods previously described [33]. Briefly, the MDA standards were prepared by dilution of 10 µL of the 4.17 M MDA standard solution with 407 µL of water to prepare a 0.1 M MDA standard solution. Further, 20 µL of the 0.1 M MDA standard solution was diluted with 980 µL of water to prepare a 2 mM MDA standard to generate 0 (blank), 0.4, 0.8, 1.2, 1.6, and 2.0 nanomole standards. Later, a sample (20 µL) of the spent holding medium was gently mixed with 500 µL of 42 mM sulfuric acid in a microcentrifuge tube. Phosphotungstic acid solution (125 µL) was added to the samples, mixed by vortex, incubated at room temperature for 5 min, and then centrifuged at 13,000× *g* for 3 min. The pellet was resuspended on ice with the 100 µL water/BHT solution (2%) and adjusted to a volume of 200 µL with water. The assay reaction was performed following the manufacturer instructions using a fluorometer (Victor 2D, Perkin Elmer, Santa Clara, CA, USA) and the data were analyzed by the software SoftMax Pro 6.5.1. (Molecular Devices, LLC, Sunnyvale, CA, USA). All samples and standards were run in duplicate.

### 2.6. Quantitative Real-Time PCR

Embryos from all groups (*n* = 24, 6 per group) were placed in 350 µL RLT buffer (Qiagen, Hilden, Germany) and stored at −80 °C. For RNA extraction from single embryos, 3.5 µL 2-mercapoethanol (Sigma-Aldrich, St. Louis, MO, USA) were added to the solution and RNA extraction and DNase I digestion were performed with the RNeasy Micro Kit (Qiagen, Hilden, Germany) according to the recommended protocol for animal and human tissues. For qPCR, 6 µL of total RNA were transcribed into cDNA using the SuperScript III First-Strand Synthesis system with random hexamer primers (Invitrogen, Carlsbad, CA, USA) in accordance with the manufacturer’s instructions. Primer and hydrolysis probes for the equine target genes *ATP1A1*, *BAX, BCL2, CYP19A1, DNMT1, DNMT3A, DNMT3B, DNMT3L, ESR1, H19, IGF1, IGF2, NANOG, POU5F1, PTGES2* and *SOX2* were designed using the PrimerQuest assay tool (https://eu.idtdna.com/PrimerQuest/Home/Index; accessed on 6 May 2021; Integrated DNA Technologies, Coralville, IA, USA) or taken from the literature [34,35,36,37]. Two reference genes (RG), *PSMB4* and *SNRPD3*, were included for normalization [38]. Assay details and full names of the genes investigated are listed in Table 1. All hydrolysis probes were dual-labelled with 6-carboxyfluorescein (FAM) on the 5′ end and Black Hole Quencher 1 (BHQ1) on the 3′ end. The assays were validated by generation of standard curves to determine PCR reaction efficiencies using the formula E = 10^−1/slope^ − 1 [39]. Efficiency-corrected C_q_ values were used for analysis. Real-time PCR quantification of the target genes using hydrolysis probes was performed as described [37]. The RGs were measured with the fluorescent DNA dye SYBR Green. Reaction conditions were described previously [38]. Target gene expression levels were normalized to the geometric mean of *PSMB4* and *SNRPD3*, and relative expression changes were calculated with the comparative 2^−ΔΔCT^ method [40].

### 2.7. Gene-Specific DNA Methylation Analysis

The analysis of gene-specific DNA methylation was based on a protocol previously implemented for bovine oocytes [41]. Briefly, DNA from single equine embryos (*n* = 28; *n* = 7 per group) was isolated and bisulfite-treated with the EZ DNA Methylation Direct Kit (Zymo Research, Irvine, CA, USA) following the recommended protocol for samples containing up to 2 × 10^3^ cells. Bisulfite-converted DNA was eluted with 10 µL M-Elution Buffer (Zymo Research, Irvine, CA, USA) into 1.5 mL DNA LoBind tubes (Eppendorf, Hamburg, Germany). Selection of target regions for bisulfite sequencing and primer design was done with the MethPrimer online tool [42]. The genomic sequences 2,000 bp upstream of the transcription start sites of the equine genes, *CYP19A1, DNMT1, DNMT3A, DNMT3B, ESR1, NANOG, PTGES2* and *SOX2* were screened for CpG islands or CG-rich regions using the CpG island prediction function of the MethPrimer tool. For multiplex nested PCR, two sets of primers, an outer primer for a first-round multiplex PCR amplification of all eight genes and an inner primer for a second gene-specific single nested PCR were designed for each gene (Table 2). Primers were designed to bind outside the CG-rich areas and to amplify as many CpG dinucleotides as possible. Only one primer set could be designed for PTGES2, which was used in both PCR reactions. The multiplex PCR was performed in 25 µL reaction volumes including 200 µM of each dNTP, 1 × buffer B2 (Solis BioDyne, Tartu, Estonia), 3 mM MgCl_2_, 120 nM of each outer primer, 1.2 units HOT FIREPol DNA polymerase (Solis BioDyne, Tartu, Estonia) and 2 µL bisufite-treated DNA. The PCR reaction was carried out using the following temperatures: initial denaturation at 95 °C for 10 min, followed by 34 cycles of 95 °C for 30 sec, 54 °C for 30 sec, and 72 °C for 1 min, and a final elongation step at 72 °C for 7 min. One µL of the multiplex PCR product was used as the template for the subsequent nested PCR performed in 25-µL reaction volumes using the same mastermix components as described for the multiplex PCR, except that the outer primer-mix was replaced with 500 nM of each gene-specific inner forward and reverse primer. The temperature protocol for multiplex PCR was also applied for the nested PCR, except that the annealing temperature and cycle number were adjusted for each gene (*CYP19A1*: 56 °C, 35 cycles; *DNMT1* and *DNMT3A*: 59 °C, 35 cycles; *DNMT3B*, *ESR1* and *NANOG*: 60 °C, 35 cycles; *PTGES2*: 56 °C, 40 cycles; *SOX2*: 58 °C, 40 cycles). Aliquots of the PCR products were run in a 2% agarose gel to confirm the correct size of the amplicons. The remaining aliquots were used for direct Sanger sequencing performed by Microsynth, Vienna, Austria. To improve the quality of the sequencing data, the amplicons of *ESR1* and *PTGES2* were isolated from the 2% agarose gel and purified using the Zymoclean Gel DNA Recovery kit (Zymo Research, Irvine, CA, USA) prior to sequencing. Data analysis was performed with CLC Genomics Workbench 9 software (Qiagen). The sequence electropherograms were investigated for methylated “C” and unmethylated “T” peaks within a CpG context. CpGs with a “C” signal > 80% were categorized as methylated, whereas methylation values < 20% were categorized as unmethylated. Methylation values between 20 and 80% were designated as an unclear methylation status [41]. 

### 2.8. Global Methylation-Immunofluorescence Staining for 5mC and 5hmC

Embryos (*n* = 28; *n* = 7 per group) from all treatment groups were prepared for immunofluorescence staining as previously described [43,44], with minor modifications. Briefly, embryos were removed from the holding medium, washed in PBS and fixed in ice-cold 4% paraformaldehyde (PFA; Sigma-Aldrich, St. Louis, MO, USA) for 25 min at room temperature (RT). Embryos were washed in PBS and kept for 25 min at RT in PBT (0.05% Tween-20 in PBS), permeabilized in 0.2% triton X-100 solution for 40 min at RT, washed three times and stored in 100 µL of PBT at 4 °C for the antibody staining. Embryos were depurinated in 4N HCl 0.1% Triton X-100 for 20 min at RT, washed in PBS and kept in PBT for 30 min at RT and incubated in blocking solution (2% BSA in PBT) overnight at 4 °C. Later, embryos were incubated with the primary antibody (5mC mouse monoclonal antibody, EpiGentek, Farmingdale, NY, USA; or 5hmC mouse monoclonal antibody, Active Motif, Carlsbad, Ca, USA) at 1:200 in blocking solution for 1h at RT, washed in PBT and incubated with Alexa Fluor 594 goat antimouse IgG (Thermo, Waltham, MA, USA) for 1h in blocking solution at RT in the dark. Finally, embryos were washed in PBT, incubated in DAPI solution for 10 min at RT, then washed in PBT and prepared in a chamber for confocal microscopy (LSM 880, Carl Zeiss, Oberkochen, Germany). Scanning was conducted with Z stack of 25 optical series from the bottom to the top of the embryos with a step size of 65 µm to allow three-dimensional distribution analysis. Images were obtained at 20 × objective magnification and analyzed using ImageJ software (version 1.50f). The fluorescence values from the embryos were recorded as integrated densities in arbitrary units (au). The following formula was used to analyze the correct relative fluorescence intensity (FI) for 5mC and 5hmC: FI = (integrated density of 5mC or 5hmC/integrated density of DAPI).

### 2.9. Statistical Analysis

The software SPSS version 24 (IBM-SPSS, Armonck, NY, USA) was used for statistical analyses. Data were tested for normal distribution by Kolmogorov-Smirnov test. Because embryo size was not normally distributed (*p* < 0.05), comparison of embryo size among groups was made with nonparametrical tests (Mann-Whitney test for fresh embryos collected on days seven and eight, Wilcoxon test for the comparison of embryo size before and after storage in groups E5C and E20C, respectively). For the analysis of pH, lipid peroxidation, gene expression, and DNA methylation, one-way ANOVA with subsequent Tukey test were used to analyze differences among groups. Spearman’s rank correlation was calculated to evaluate the relationships among the genes. Data are shown as mean ± standard error of the mean (SEM). A *p*-value < 0.05 was considered statistically significant.

## 3. Results

### 3.1. Recovery Rate and Embryo Morphology

A total of 80 embryos were obtained from 144 embryo flushing procedures; therefore, the overall recovery rate was 55% (80/144). Of the embryos collected, 92% were at the blastocyst and 8% at the morula stage. All embryos had a morphological classification of 1 or 2. Embryo size differed (*p* < 0.05) between fresh embryos collected on day seven (E7F; *n* = 20) and day eight (E8F; *n* = 16; Figure 1). The size of embryos stored at 5 °C (E5C; *n* = 20) was larger (*p* < 0.01), whereas in embryos stored at 20 °C (E20C; *n* = 19) the size was smaller (*p* < 0.05) after 24 h. In five embryos (one of E20C and four of E8F), data from size determination were not available. In the E5C group, there was an increase in size in 17 and a decrease in three embryos, whereas in the E20C group, there was in increase in size in four, a decrease in 12 and no change in size in three embryos (*p* = 0.02). No significant difference in size among embryos collected on day seven was detected. Fresh embryos collected on day eight were larger (*p* < 0.01) than embryos stored for 24h, irrespective of storage temperature. In 10 embryos (six in E20C and four in E5C) of the storage groups, a shrinkage-like morphology was detected irrespective of temperature (Figure 2). For the determination of gene expression and gene-specific DNA methylation, only blastocysts of similar development stage and ≥300 µm in diameter at collection (in total *n* = 52) were used.

### 3.2. Medium Temperature, pH, and Lipid Peroxidation

The temperature of the storage medium was constant for all embryos within the temperature groups during the storage period. The pH of the medium (7.22 ± 0.07 and 7.22 ± 0.09) was similar in groups E5C and E20C. In the spent holding medium, MDA accumulation could not be detected irrespective of treatment group.

### 3.3. Gene Expression

The relative mRNA abundance of specific genes related to growth and development, embryo-maternal communication, methylation and apoptosis is depicted in Figure 3. The gene *DNMTL3L* was not expressed in equine embryos irrespective of group. The expression of *IGF1, POU5F1, SOX2, NANOG, CYP19A1, PTGS2, DNMT1, DNMT3a, DNMT3b*, and *BAX* differed among groups (*p* < 0.05). Gene expression of *IGF1*, *CYP19A1* was similar in the two groups of stored embryos irrespective of temperature and control embryos collected on day seven, but higher in embryos collected on day eight. A different pattern with regard to mRNA abundance was determined for *SOX2, NANOG, PTGES2, DNMT1* and *DNMT3b*. On the one hand, the gene expression was similar in day seven control embryos and embryos stored at 5 °C and, on the other hand, day eight controls and embryos stored at 20 °C. Gene expression of *POU5F1* differed in E5C embryos in comparison to E20C and E8F groups. The expression of *DNMT3a* of E5C differed from all the other groups. *BAX* expression was lower in fresh embryos collected on day eight only when compared to stored embryos at 5 °C. Independently of the treatment groups, we observed variations in the coexpression among the specific genes related to growth and development, embryo-maternal recognition, methylation and apoptosis (Table 3).

### 3.4. Gene-Specific DNA Methylation

The percentage of overall DNA methylation and the overview of DNA methylation results of single CpGs of the specific genes are depicted in Table 4 and Figure 4, respectively. The predicted promoter region of *CYP19A1* was fully methylated whereas *PTGES2* and *SOX2* were unmethylated in all embryos irrespective of treatment group. For *DNMT1*, the methylation status was similar in the two groups of fresh embryos and embryos stored at 5 °C, whereas it was lower in embryos stored at 20 °C. The methylation status of *NANOG*, however, was higher in fresh day eight embryos than in the other three groups. 

### 3.5. Global methylation–5hmC and 5mC

The fluorescence intensity (255 to 4.97 × 10^6^ au) varied among embryos. The global methylation was, however, not affected by treatment, irrespective of the antibody (5mC or 5hmC; Figure 5). When immune expression was compared between antibodies within each treatment, E7F had a greater (*p* < 0.05) expression of 5hmC compared to 5mC. Among all other groups, expression did not differ between 5hmC compared to 5mC.

## 4. Discussion

To the best of our knowledge, the present study is the first to analyze the effects of storage temperature (5 and 20 °C) on development, relative mRNA abundance and DNA methylation in equine embryos processed for shipment using a commercial holding medium. Untreated fresh embryos collected on day seven and day eight were included as controls to determine changes in relative mRNA abundance and DNA methylation associated with embryo age itself. Whereas difference in size of embryos collected on days seven and eight after ovulation, as well as a considerable variance in size of embryos collected on the same day, is not surprising and in agreement with previous studies [45,46,47] In this context, it has to be considered that in this study ovulation was assessed at 24 h-intervals, which has most likely contributed to the variation in size within the embryos collected on days seven and eight after ovulation. Nevertheless, effects of storage temperature on equine embryo size have not yet been reported. Interestingly, embryonic size was slightly, but significantly, smaller in embryos stored at 20 °C, whereas embryos stored at 5 °C were larger after 24 h. It has been reported previously that the rapid increase in size of equine embryos starting on day seven mainly depends on influx of fluid into the blastocoel allowed by the formation of an osmotic gradient due to activity of α1/β1 Na+/K+-ATPase (reviewed by [47]). This enzyme has been detected in horse embryos not earlier than day eight after ovulation [48]. The present results suggest that storage at 20 °C may inhibit the activity of this enzyme but, in comparison to in vivo-produced embryos, decreased enzyme activity is also present in embryos stored at 5 °C. Interestingly, in all embryos analyzed for gene expression in the present study, *ATP1A1*, the gene encoding for the α1 subunit of the Na+/K+-ATPase was detected but there were no differences in relative mRNA abundance among groups. The mRNA abundance is, however, not necessarily associated with enzyme activity. In day eight embryos, immunohistochemistry of Na+/K+-ATPase revealed a pronounced protein expression of this enzyme in the whole trophoblast [49]. This suggests that the technique would not allow detection of differences among groups in the present investigation. We did not detect changes in pH and oxidative balance of the holding medium during storage irrespective of temperature. However, in approximately 25% of the embryos a partial separation of the trophoblast from the zona pellucida occurred. This finding is in accordance with a previous description of morphological abnormalities after 12 or 24 h of cooled storage in equine embryos [32] and may also be associated with delayed formation of an osmotic gradient due to impaired activity of α1/β1 Na+/K+-ATPase and subsequent collapse of the blastocoel. Among the genes we assessed in this study, *SOX2*, *NANOG* and *DNMT3B* are associated with pluripotency, and have been detected in equine pluripotent cells and in equine MSCs [50]. Interestingly, relative mRNA abundance of these genes had a positive correlation among each other in our study; the interaction among *SOX2* and *NANOG* is necessary to regulate embryonic stem cell self-renewal [51]. Notwithstanding, the relative mRNA abundance of *SOX2*, *NANOG* and *DNMT3B* was affected by storage temperature in a way that it was similar in day seven control embryos and embryos stored at 5 °C on the one hand, and day eight controls and embryos stored at 20 °C on the other hand. Changes in *PTGES2* and *DNMT1* relative mRNA abundance followed the same pattern. Moreover, correlation analysis demonstrated that *POU5F1* and *CYP19A1* have an opposed interaction at this embryo stage. This suggests that storage of equine day seven embryos at 20 °C does not prevent changes in relative mRNA abundance that also occur between day seven and day eight in embryos in utero. Consequently, a storage temperature of 20 °C may be beneficial for embryo development when compared to a storage temperature of 5 °C where gene expression of the respective candidate genes stayed at the same level as in the control embryos collected on day seven. The relative mRNA abundance of other candidate genes, namely *IGF1*, a gene associated with embryonic development because it increases cell proliferation and decreases cell apoptosis [52,53], *CYP19A1*, the gene encoding aromatase, and the proapoptotic gene *BAX*, was similar among stored embryos irrespective of temperature, as well as in control embryos collected on day seven, but differed in embryos collected on day eight. This suggests that embryo storage at 20 °C may have beneficial effects in comparison to 5 °C regarding the expression of some, but not all, genes. These findings are in agreement with higher pregnancy rates after transfer of shipped equine embryos with an arrival temperature between 10 and 16 °C in comparison to embryos transported in cooler conditions [54].

Methylation of DNA occurs during the migration of proliferating primordial germ cells but demethylation in postmigratory germ cells. A second wave of DNA demethylation takes place in cleavage stage embryos, with DNA methylation being minimal at the blastocyst stage ([55]; recently reviewed by [56]). In the present study, only minor differences in the methylation status of candidate genes were detected between fresh embryos collected on day seven and day eight, and the two groups of stored embryos. An interesting finding was hypermethylation of *NANOG* in embryos collected on day eight in comparison to the other groups, demonstrating that *NANOG* methylation reprogramming is highly active between days seven and eight of equine embryonic development. A hypomethylation of *DNMT3a* in fresh day-seven embryos in comparison to all the other groups where embryo age was eight days, is most probably only age-related. Changes in the methylation status of genes that regulate embryo development have been described in porcine blastocysts during in vitro culture and were suggested to contribute to early pregnancy loss [57]. In the present study, either demethylation or a delay in methylation of CpG islands of specific genes could be a consequence of maintaining transcriptional activation [58] at the temperatures used during storage compared to physiological temperatures. The embryo environment associated with ART had long-lasting consequences in mouse, ruminant and human embryos [59,60,61]. A major setback associated with the introduction of in vitro production of ruminant embryos was the occurrence of large offspring syndrome [59]. This condition has caused considerable loss after transfer of in vitro produced ruminant embryos and has been linked to the use of serum in the culture media [59] altering gene expression and DNA methylation [14]. Comparable problems, however, have not yet been described in foals derived by ART, which is in agreement with the findings of the present study with only minor changes of methylation status of the candidate genes.

## 5. Conclusions

Our study investigated for the first time the relative mRNA abundance of some candidate genes, as well as the global and gene-specific DNA methylation status in fresh equine embryos collected on days seven and eight after ovulation. Moreover, we demonstrate that short-term storage of embryos alters the expression of genes involved in embryo development and methylation. Results suggest some beneficial effects of storage at 20 °C in comparison to 5 °C. However, further studies to clarify impacts after embryo transfer on implantation, embryo development and pregnancy success will benefit the future of ARTs in equine industry.

## Figures and Tables

**Figure 1 animals-11-01325-f001:**
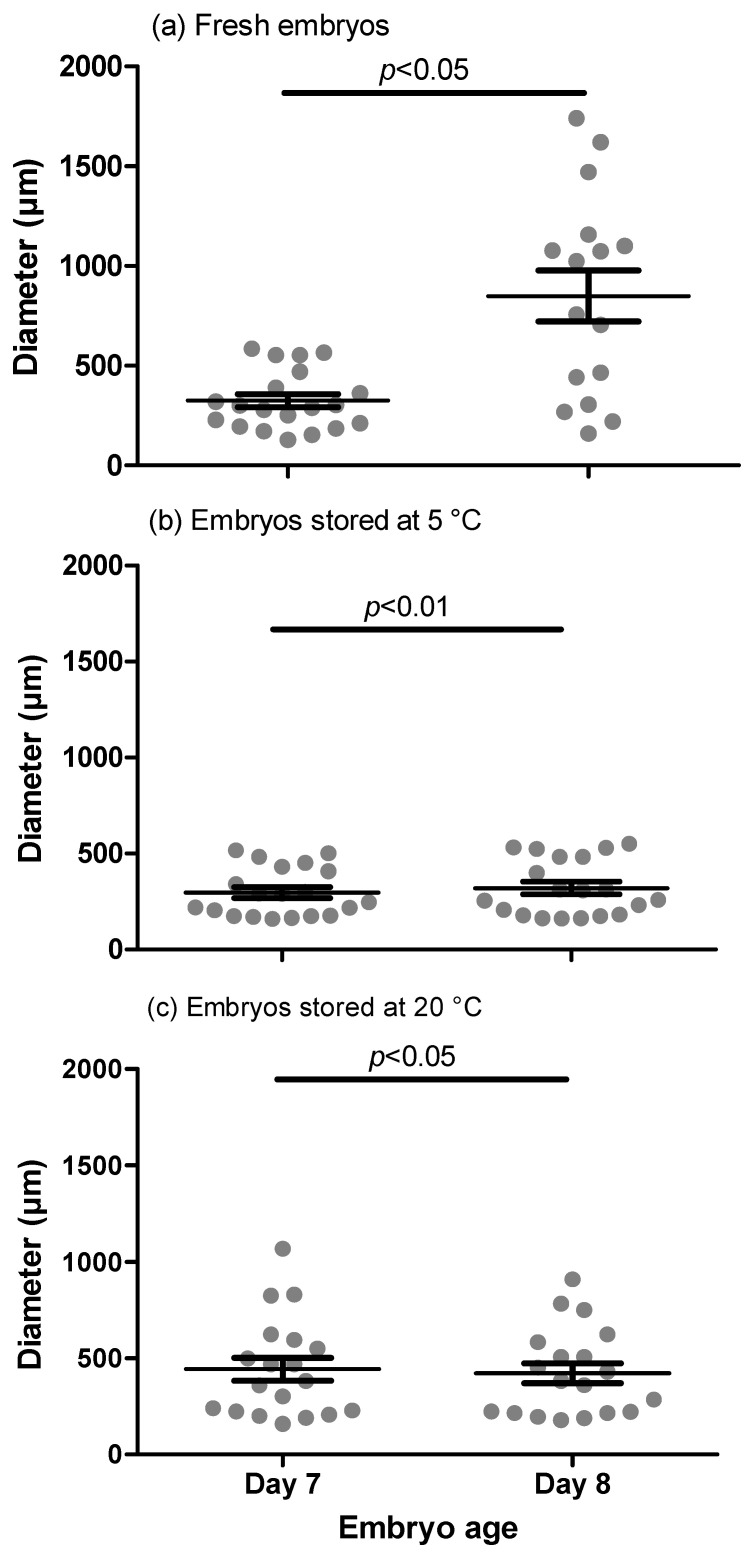
Diameter of fresh embryos. (**a**) collected on day seven (E7F) and day eight (E8F) after ovulation, and before (d7) and after 24 h (d8) short-term storage at (**b**) 5 °C (E5C) and (**c**) 20 °C (E20C). Each dot represents a single embryo. The plot represents the descriptive variation (mean ± S.E.M.) within the group. Significant differences are indicated in the figure.

**Figure 2 animals-11-01325-f002:**
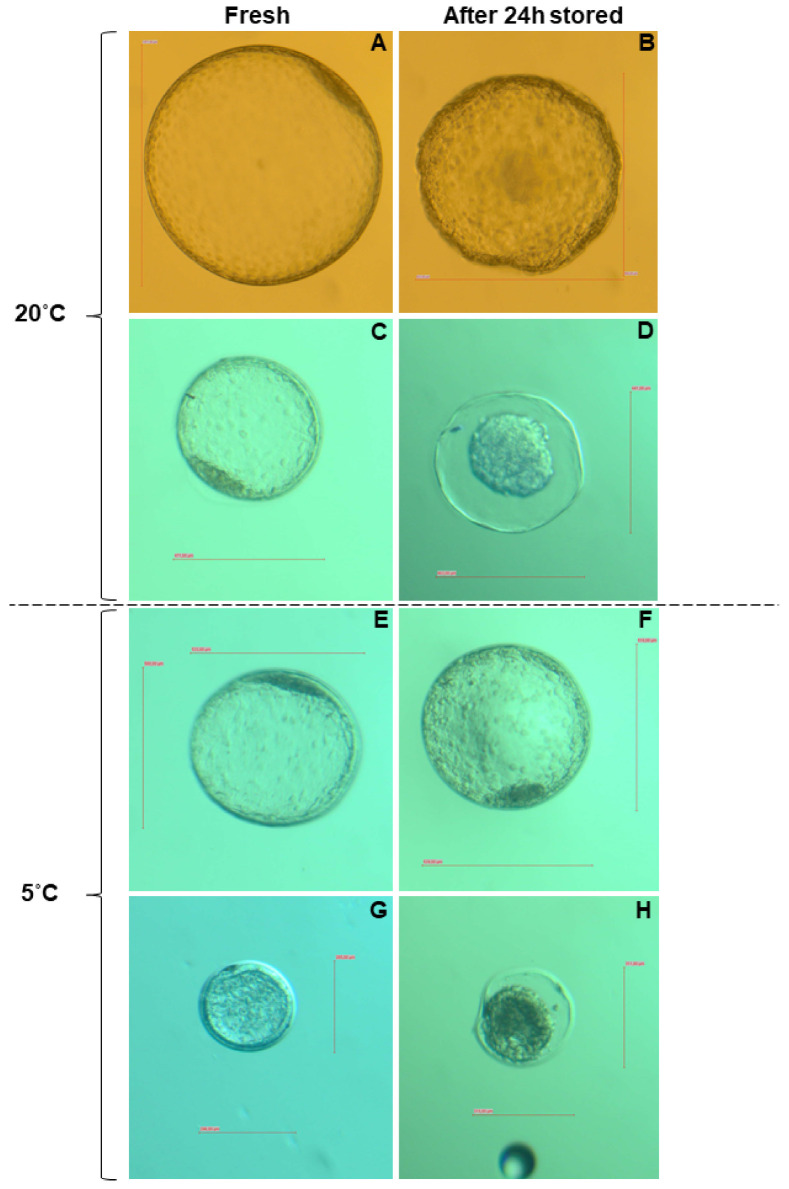
Illustration of fresh equine embryos (E7F) and stored in either 20 °C or 5 °C for 24h. (**A**,**C**,**E**,**G**) Embryos at blastocyst stage with normal morphology before submission to storage and (**B**,**D**,**F**,**H**) the same embryos after storage with (**B**) embryo with a slight shrinkage of the embryonic cells and few areas of detachment between the trophoblast cells, zona pellucida and embryonic capsule; (**F**) embryo with normal morphology after storage time; (**D**,**H**) embryos with strong shrinkage morphology of the embryonic cells.

**Figure 3 animals-11-01325-f003:**
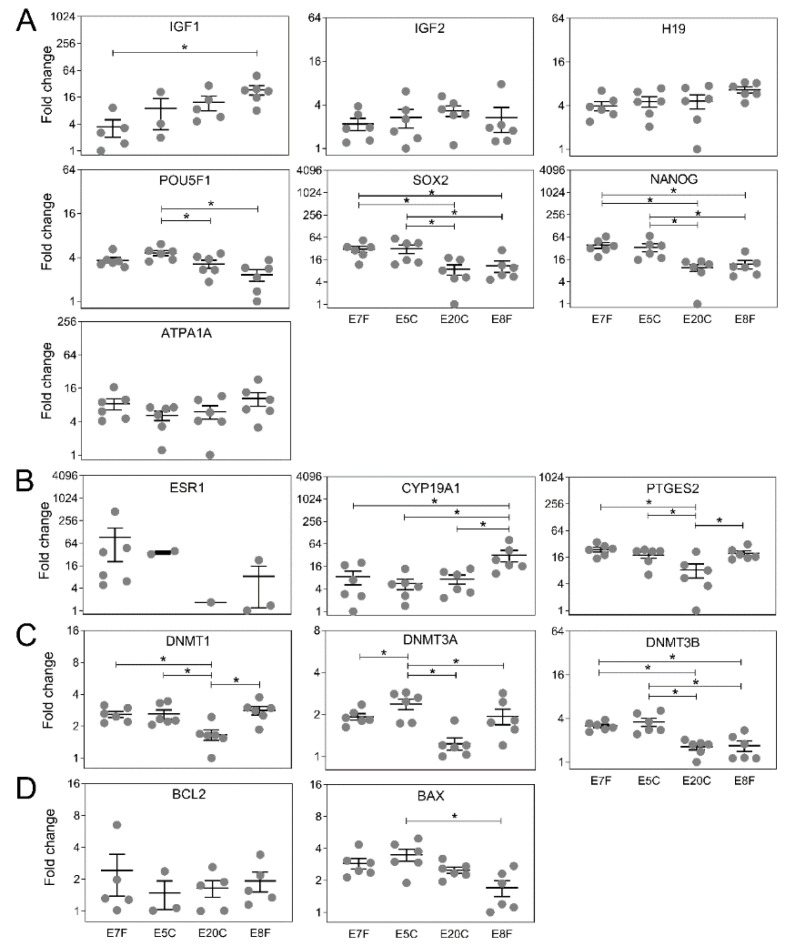
Relative mRNA expression levels of genes related to: (**A**) growth and development, (**B**) embryo-maternal communication, (**C**) methylation, and (**D**) apoptosis in fresh control (E7F and E8F) and stored embryos (E5C and E20C). E5C and E20C either did not express ESR1 or did not have enough samples; therefore, values were not considered for statistical comparison. Each dot represents a single embryo. The plot represents the descriptive variation (mean ± S.E.M., *n* = 6) within the group. Fold-changes were calculated with the comparative 2^−ΔΔCT^ method (* *p* < 0.05).

**Figure 4 animals-11-01325-f004:**
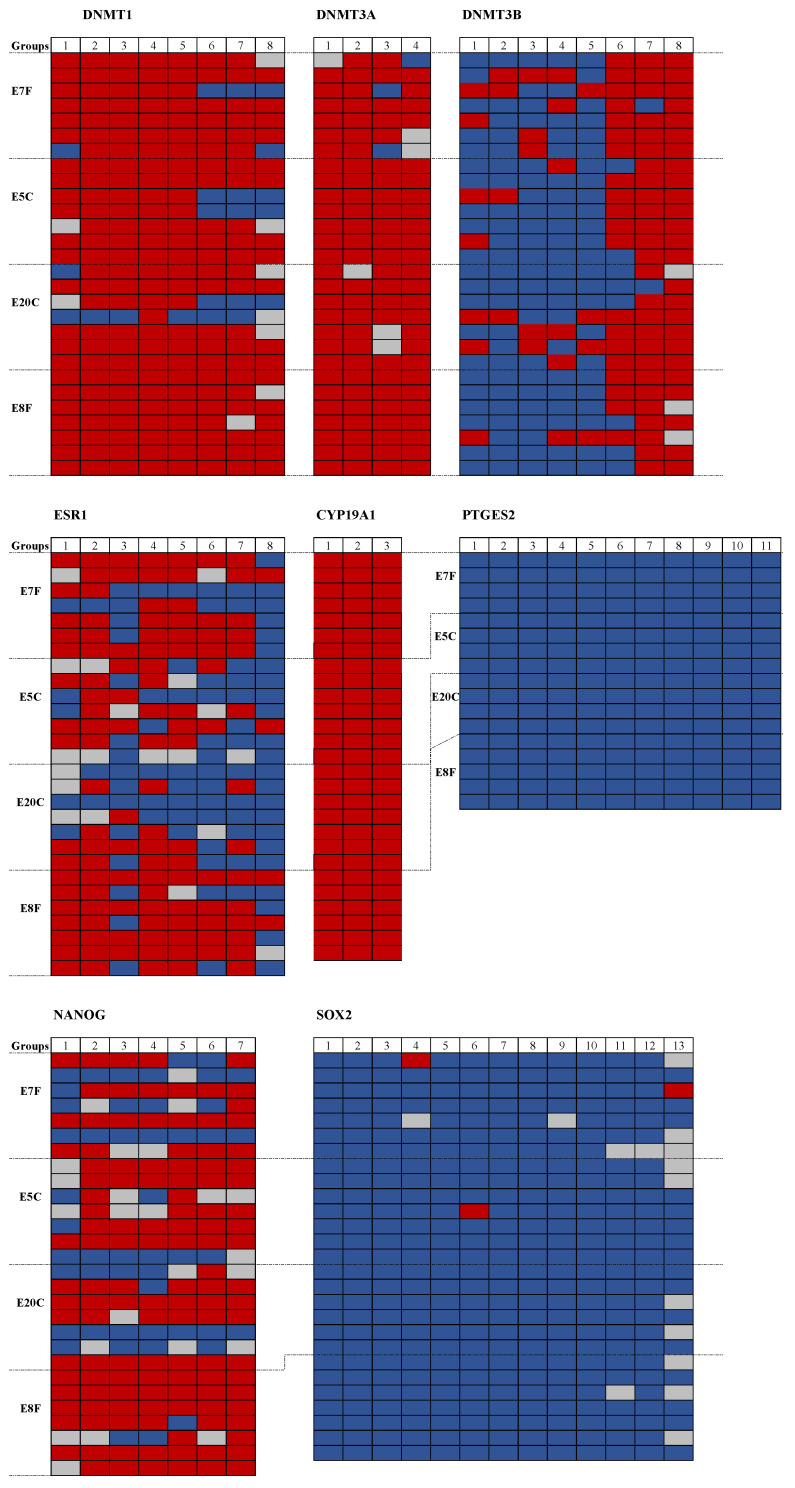
Overview of DNA methylation results of single CpGs for the selected genes. DNA methylation (*DNMT1, DNMT3A, DNMT3B*), embryo-maternal recognition (*ESR1, CYP19A1, PTGES2*), and growth and development (*NANOG, SOX2*) of the equine embryo. Blue box, unmethylated CpG; red box, methylated CpG; gray box, unclear methylation state. One row refers to one embryo and the number of columns refers to the number of CpGs sequenced in the CG-rich region (methprimer CpG island finder) upstream of the gene (Table 2).

**Figure 5 animals-11-01325-f005:**
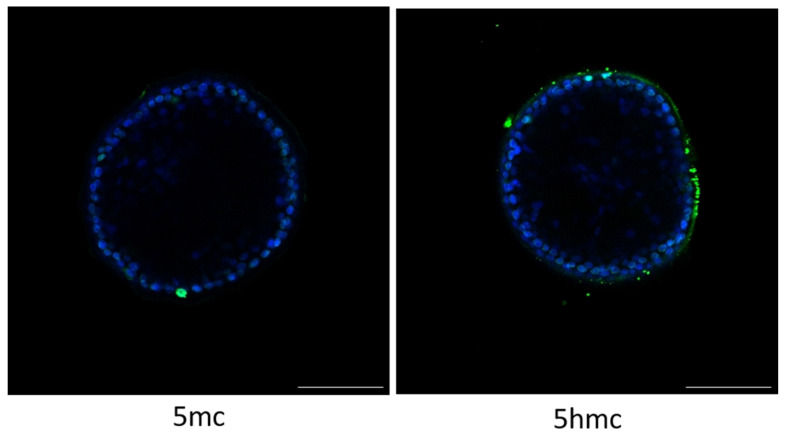
Illustrative image of 5mC and 5hmC immunolabeling in equine stored embryo at 20 °C after 24 h. Equine embryos had similar 5mC and 5hmC immunolabeling among treatments. Green labeling represents methylated stained cells. Blue labeled cells are stained with DAPI to allow the identification of the embryonic cells and permit the quantification of global methylation. Scale bar, 100 µm.

**Table 1 animals-11-01325-t001:** Primer for quantitative PCR.

Gene Symbol	Gene Name	NCBI/Ensembl Accession Number	Oligo Sequence (5′–3′)	Amplicon Length (bp)	PCR Efficiency (%)	R^2^ Value	Reference
ATP1A1	Equus caballus ATPase Na+/K+ transporting subunit alpha 1	NM_001114532.2, XM_023640223.1, XM_023640224.1	F: CTTGATGAACTTCAGCGCAAATA	104	94	0.990	This study
			R: GGTGTAAGGGCATTGGGA				
			P: TGAGCCGAGGCTTAACAACTGCTC				
BAX	Equus caballus BCL2-associated X protein	XM_014729721.1	F: AGGATGCGTCCACCAAGAAG	80	93.2	0.994	[37]
		XM_014729717.1	R: CCTCTGCAGCTCCATGTTACTG				
			P: CTCAAGCGCATCGGAGATGAGCTG				
BCL2	Equus caballus B-cell lymphoma 2	XM_001490436.2	F: TTGGAAAGCCTACCACTAATTGC	74	92.6	0.998	[37]
			R: CCGTGTTTATAGGCACAGGAGAT				
			P: CCCACCTGAGCGGCTCCACC				
CYP19A1	Equus caballus cytochrome P450 family 19 subfamily A member 1	NM_001081805.2	F: GGAGAGGAAACGCTCGTTATTA	107	99.2	0.999	This study
		XM_005602588.2	R: CCCATATACTGCAACCCAAATG				
		XM_005602587.2	P: ATCACTACTCCTCCCGATTTGGCA				
DNMT1	Equus caballus DNA methyltransferase 1	XM_014741825.1	F: GACCACCATCACGTCTCATTT	97	100.5	1	This study
			R: CTCCTCATCCACAGAATTGTCC				
			P: AAACGGAAACCCGAGGAAGAGCTG				
DNMT3A	Equus caballus DNA methyltransferase 3 α	XM_005600169.2	F: GATTATTGACGAACGCACAAGAG	112	100	0.998	This study
		XM_005600168.2					
		XM_005600167.2					
		XM_005600170.2	R: GTGTTCCAGGGTGACATTGA				
		XM_005600171.2	P: TGCAAATGTCTTCGATGTTCCGGC				
DNMT3B	Equus caballus DNA methyltransferase 3 β	XM_001916514.4	F: CGAGTCTTGTCCCTGTTTGAT	110	100.6	0.999	This study
			R: GCGATAGACTCTTCACACACTT				
			P: CGCCACAGGGTACTTGGTTCTCAA				
DNMT3L	Equus caballus DNA (cytosine-5-)-methyltransferase 3-like	XM_014736476.1	F: GCCCTCACTTGGTTGGTTT	98	100.5	0.999	This study
			R: CTTCCACACAGGCACAGTTT				
			P: CAAAGTGCCCATCTGCTCTGGAGA				
ESR1	Equus caballus estrogen receptor 1	NM_001081772.1	F: CACCCAGGAAAGCTCCTATTT	110	101.8	0.999	This study
			R: CGAGATGACGTAGCCAACAA				
			P: TCCACCATGCCCTCTACACATTTCC				
H19	Equus caballus H19, imprinted maternally expressed transcript	NR_027326.2	F: CCTCTAGCTCTGACTCAAGAATATG	103	94.3	0.992	This study
			R: CAGGTCCATCTGGTTCCTTTAG				
			P: ACTCAGGAATCAGCTCTGGAAGGT				
IGF1	Equus caballus insulin-like growth factor 1	NM_001082498.2	F: TGCTTCCGGAGCTGTGATCT	67	102	1	[34]
		XM_005606471.2					
		XM_005606472.2					
		XM_005606470.2	R: CCGACTTGGCAGGCTTGA				
		XM_005606469.2	P: AGGAGGCTGGAGATGTACTGCGCACC				
IGF2	Equus caballus insulin-like growth factor 2	NM_001114539.2	F: AAGTCCGAGAGGGACGTG	100	99.9	0.998	This study
			R: ATTGCTTCCAGGCGTTGT				
			P: CCCGTGGTCAAGCTCTTCCAGT				
NANOG	Equus caballus Nanog homeobox	XM_014740545.1	F: ACAGCCCCGATTCATCCA	72	102.3	0.999	[35]
			R: TCTTTGCCTCGCTCGTCTCT				
		XM_001498808.1	P: CAGTCCCAGAGTAAAACCGCTGCCC				
POU5F1	Equus caballus POU class 5 homeobox 1	XM_014734675.1	F: CGGGCACTGCAGGAACAT	73	100.8	0.999	[35]
			R: CCGAAAGAGAAAGCGAACTAGTATTG				
		XM_001490108.5	P: TTCTCCAGGTTGCCTCTCACTCGGTTC				
PSMB4	Equus caballus proteasome subunit beta type IV	XM_001492317.4	F: CTTGGTGTAGCCTATGAAGCCC	82	93.1	0.991	[38]
		XM_005610132.1					
		XM_008515015.1					
		XM_005613704.1	R: CCAGAATTTCTCGCAGCAGAG				
PTGES2	Equus caballus prostaglandin-endoperoxide synthase 2	NM_001081775.2	F: GAGGTGTATCCGCCCACAGT	81	92.3	0.996	[36]
			R: AGCAAACCGCAGGTGCTC				
			P: TCAGATGGAAATGATCTACCCGCCTCA				
SNRPD3	Equus caballus small nuclear ribonucleoprotein D3 polypeptide.	XM_001489060.4	F: ACGCACCTATGTTAAAGAGCATG	120	99.4	0.996	[38]
		XM_008511652.1	R: CACGTCCCATTCCACGTC				
SOX2	Equus caballus SRY box 2	XM_003363345.3	F: TGCGAGCGCTGCACAT	91	99.3	0.998	[35]
			R: AGCGTGTACTTATCCTTCTTCATGAG				
			P: ATAAATACCGTCCTCGGCGGAAAACCAA				

R^2^: correlation coefficient of standard curve; F and R: forward and reverse primer; P: hydrolysis probe.

**Table 2 animals-11-01325-t002:** Multiplex nested PCR primer for DNA methylation analysis.

Gene	Primer Sequence (5′–3′)	Amplicon Length (bp)	Genomic Localization (EquCab3.0)	Number of CpGs in Inner Amplicon	Number of Sequenced CpGs
CYP19A1	Outer forward: TTTTAGTTTTGATTGGTTGTTTTT	317	1:140151317-140151633	-	-
	Outer reverse: CTAAACCCCATAAAACATCTCTTAC				
	Inner forward: TTTTTTTTGTAAGATTAGTGAGTATATTTA	212	1:140151386-140151597	4	3
	Inner reverse: TTTCCAAAATTAAAAAACATAACC				
DNMT1	Outer forward: AATTTTTTTTAAGAGTTTGGTATGG	264	7:51536766-51537029	-	-
	Outer reverse: ACCAATCCTCCTCTTTATACTAAAA				
	Inner forward: GAGTTTGGTATGGTATATAAGTGTTGA	229	7:51536778-51537006	9	8
	Inner reverse: AAAAAACTAACCCTAAACTCACATC				
DNMT3A	Outer forward: GGGATTGATTAGATTTTTTAGAGAAG	316	15:71656615-71656930	-	-
	Outer reverse: TAATAACACTAAATCCCTCCAAAAC				
	Inner forward: TAGGAGTTTAGTGGGGGAATAGT	200	15:71656665-71656864	6	4
	Inner Reverse: ATAAAATAAATAAAACCCCTACACC				
DNMT3B	Outer forward: TTAAAGGGGGAATAGTAGAAGTTTA	388	22:24214245-24214632	-	-
	Outer reverse: CAACTCCAAAAATATTTAAAATCAC				
	Inner forward: TATAGAGGATGGATTTGGGATTTTA	240	22:24214309-24214548	10	8
	Inner reverse: ACTAAACACTCCCTACCCTAATACC				
ESR1	Outer forward: TTGTGGTAGGTATGAATATTTATGTG	334	31:15363114-15363447	-	-
	Outer reverse: ATTACATATACAACCAACCACAAAC				
	Inner forward: AATTTTTAGTGGGAGGAAGTATAGTAT	226	31:15363156-15363381	9	8
	Inner reverse: ACATAAACTAACAAAAAACATCCC				
NANOG	Outer forward: TGGAAATATGGTGAATTTATAGGTAT	387	6:36542573-36542959	-	-
	Outer reverse: AACTTAAATATCCAAACAAAAAACC				
	Inner forward: TTGGTAGATAGGATTAATTGAGAATT	237	6:36542585-36542821	8	7
	Inner reverse: CAAACAAAAAACCTTAAAAAAATAC				
PTGES2	Forward: GATTTATTTAAGAGTGGGGGAGGT	205	25:32059289-32059493	17	11
	Reverse: CAATATAAAACCCCAACC				
SOX2	Outer forward: ATTTTTAATATAGAATAAATTATGGAGAAG	302	19:22733114-22733417	-	-
	Outer reverse: AAATAAAAATAAAACAAAACAAAATAAATA				
	Inner forward: ATAGAATAAATTATGGAGAAGTAAGGAG	253	19:22733122-22733376	19	13
	Inner reverse: CTATCCTACTAAAATTTCAAAAACC				

**Table 3 animals-11-01325-t003:** Spearman rank correlation coefficient (Spearman’s rho) among the specific genes in equine embryos.

Function		Growth & Development							Embryo-Maternal Communication			Methylation			Apoptosis	
	Gene	IGF2-	H19	POU5F1	SOX2	IGF1	NANOG	ATPA1A	ESR1	CYP19A1	PTGES2	DNMT1	DNMT3a	DNMT3b	BCL2	BAX
Growth & Development	IGF2	1														
	H19	0.4 *	1													
	POU5F1	0.26	−0.05	1												
	SOX2	0.21	−0.14	0.74 ***	1											
	IGF1	−0.06	0.34	0.74 ***	−0.66 **	1										
	NANOG	0.07	−0.27	0.73 ***	0.89 ***	−0.63 **	1									
	ATPA1A	−0.04	0.06	−0.66 ***	−0.46 *	0.56 *	−0.34	1								
Embryo-maternal communication	ESR1	0.04	−0.05	0.33	0.45	−0.43	0.2	0.18	1							
	CYP19A1	−0.24	0.4 *	−0.74 ***	-0.59 **	0.49 *	−0.51 *	0.5 *	−0.12	1						
	PTGES2	0.14	0.16	−0.1	0.14	0.03	0.26	0.37	0.15	0.1	1					
Methylation	DNMT1	−0.23	0.29	−0.11	0.1	0.49 *	0.19	0.45 *	0.27	0.47 *	0.7 ***	1				
	DNMT3a	−0.12	0.19	0.32	0.5	0.02	0.58 **	0.14	0.54	−0.04	0.45 *	0.65 ***	1			
	DNMT3b	−0.04	−0.05	0.75 ***	0.82 ***	−0.59 **	0.91 ***	−0.36	0.29	−0.39	0.3	0.29	0.57 **	1		
Apoptosis	BCL2	−0.04	0.24	0.21	0.05	0.01	0.14	0.03	0.38	0.09	−0.07	0.23	0.43	0.07	1	
	BAX	0.07	−0.22	0.77 ***	0.55 **	−0.69 **	0.58 **	−0.6 **	0.34	−0.54 **	−0.04	−0.12	0.05	0.67 ***	−0.02	1

Color gradient to indicate where each Spearman correlation coefficient value falls within the range, blue = −1 to red = 1 R value. Statistical significance for the correlation coefficient is represented as * *p* < 0.05, ** *p* < 0.01, *** *p* < 0.001.

**Table 4 animals-11-01325-t004:** Percentage of methylation in the CpG islands of the specific genes ^†^.

Function	Gene	E7F	E5C	E20C	E8F	*p*-Value
Growth & Development	SOX2	2.2	1.1	0.0	0.0	0.62
	NANOG	46.9 ^a^	63.3 ^a,b^	53.1 ^a^	85.7 ^b^	<0.0001
Embryo-maternal communication	ESR1	64.3 ^a,b^	39.3 ^b,c^	28.6 ^c^	78.6 ^a^	<0.0001
	CYP19A1	100.0	100.0	100.0	100.0	NS
	PTGES2	0.0	0.0	0.0	0.0	NS
Methylation	DNMT1	89.3 ^a,b^	85.7 ^a,b^	75.0 ^b^	96.4 ^a^	0.009
	DNMT3a	78.6 ^a^	100.0 ^b^	89.3 ^a,b^	100.0 ^b^	0.006
	DNMT3b	53.6	30.4	44.6	33.9	0.052

^†^ Specific genes related to growth & development, embryo-maternal recognition, and methylation functions in fresh control (E7F and E8F) and stored embryos (E5C and E20C); embryos day seven fresh, E7F; stored at 5 °C, E5C, and 20 °C, E20C; fresh embryos day eight, E8F. ^a,b,c^ Differences among groups within genes are indicated by superscript letters and are highlighted in gray (*p* < 0.05).

## Data Availability

The data analyzed during the current study are available from the corresponding authors on reasonable request.

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
