# Peer review of "Epigenetic Changes in Equine Embryos after Short-Term Storage at Different Temperatures"

_animals, 2021, doi:10.3390/ani11051325_

Round 1

Reviewer 1 Report

The aim of this study is the comparison of different storage temperatures on epigenetic changes in equine embryos. The comparison of stored embryos with fresh embryos, which revealed even more differences, could be specified as additional aim in the abstract / simple summary. Many of the investigated parameters showed only minor differences between the two temperatures, but more alterations between fresh and stored embryos.

The abstract is in some extent not consistent with the findings that were discussed in detail (e.g. ESR1, NANOG, DNMT1 hypomethylation).

Also the conclusion, that the results give reason to assume that the 20°C temperature might be beneficial should be mentioned in these first two sections. In the abstract it sounds, that there is no temperature effect at all. But in the conclusion, the 20°C are claimed to be beneficial.

Also, the used medium is important and should be mentioned in the abstract already.

Material and Methods:

A critical point in the experimental design is to check for ovulation only every 24 hours. On day 0 (ovulation detection) it’s possible, that the ovulation had already occurred almost 24 hours before. So, it’s difficult to analyze differences between day 7 and day 8. There was no significant difference between the size of embryos flushed on day 7 (E7F, E5C, E20C)? In figure 1 day 7 embryos of E20C includes a couple of very large embryos and the mean (?) seems to be visible higher. And on the other hand, there is no difference at all visible in the graph for the significant differences in the E5C and E20C group.

The huge variability in the E8F group could easily be the result of the possible ovulation window of 24 hours.

Line 150:         Please specify the holding medium correctly. EquiHold?

Results:

Line 327: “No significant difference in size among embryos collected on day 7 was detected.”

-> Can you provide the data for this? Diameter, Means, SD, p-value?

Line 333: “For the determination of gene expression and DNA methylation, only blastocysts of similar development stage and ≥300 μm in diameter at collection (in total n = 52) were used.”

-> This needs to be explained in the Material and Methods section, including an update of the numbers:

2.6.      qPCR                                                  n=24, 6 per group

2.7.      gene-specific DNA methylation         n=28, 7 per group

2.8.      global methylation                            n=28, 7 per group

Please update the correct numbers, also in the Results section.

Figure 1: Description on the plotted parameters (mean/error/CI) and statistical tests are missing in the figure legend. For b and c the diameter axis could be shorten to visualize the significant differences. Can you add a figure for E8F versus E5C / E20C on d8? To illustrate the restricted growth of the storage embryos?

Can you mark the embryos < 300 mm in diameter that were not used for gene expression analysis and DNA methylation analysis?

Figure 2: Are these the same embryos: A/B, C,D, E/F, G/H?

Figure 3: Which values where plotted (mean / error / CI)?

Discussion:

Line 427: “The present results suggest that storage at 20 °C inhibits the activity of this enzyme, but in comparison to in vivo-produced embryos decreased enzyme activity is present in embryos stored at 5 °C.”

This sentence is slightly confusing… both, 5°C and 20°C embryos were significantly smaller on day 8 than in vivo derived embryos? Can you quantify the size differences between the groups (see comment for Figure 1). And better explain your suggestion for the difference between 5° and 20°?

Author Response

Response to Reviewer 1 comments.

The aim of this study is the comparison of different storage temperatures on epigenetic changes in equine embryos. The comparison of stored embryos with fresh embryos, which revealed even more differences, could be specified as additional aim in the abstract / simple summary. Many of the investigated parameters showed only minor differences between the two temperatures, but more alterations between fresh and stored embryos.

Authors’ response: We thank the reviewer for this comment. Certainly, our goal was to evaluate different storage temperatures on epigenetic changes in equine embryos. The fresh embryos served as control groups. Therefore, the differences encountered between stored and fresh embryos revealed the overall storage effects with regard to epigenetic changes.
--

The abstract is in some extent not consistent with the findings that were discussed in detail (e.g. ESR1, NANOG, DNMT1 hypomethylation).

Authors’ response: We thank the reviewer for this comment. We have checked these statements and rewritten the sentence to avoid confusion (Percentage of methylation in the CpG islands was lower in the specific genes ESR1, NANOG, and DNMT1 (P<0.001) in E20C embryos when compared to E8F (advanced embryo stage).
--

Also the conclusion, that the results give reason to assume that the 20°C temperature might be beneficial should be mentioned in these first two sections. In the abstract it sounds, that there is no temperature effect at all. But in the conclusion, the 20°C are claimed to be beneficial.

Authors’ response: Thanks for noticing it. We have added this to the abstract conclusion (lines 57-58) to match with the general conclusion as follows:

“Although our results suggest some beneficial effects of storage at 20 °C in comparison to 5 °C, the short-term storage, regardless of temperature, modified gene expression and methylation of genes involved in embryo development and may hence compromise embryo viability and development after transfer.”
--

Also, the used medium is important and should be mentioned in the abstract already.

Authors’ response: Thanks. We have added the name of the medium to the abstract (Line 42).
--

Material and Methods:

A critical point in the experimental design is to check for ovulation only every 24 hours. On day 0 (ovulation detection) it’s possible, that the ovulation had already occurred almost 24 hours before. So, it’s difficult to analyze differences between day 7 and day 8. There was no significant difference between the size of embryos flushed on day 7 (E7F, E5C, E20C)? In figure 1 day 7 embryos of E20C includes a couple of very large embryos and the mean (?) seems to be visible higher. And on the other hand, there is no difference at all visible in the graph for the significant differences in the E5C and E20C group.

The huge variability in the E8F group could easily be the result of the possible ovulation window of 24 hours.

Authors’ response: We thank the reviewer for bringing up this point of criticism. This is a certain limitation of the present study. On the one hand, there is a clear difference with regard to the mean size of embryos collected on days 7 and 8 although some of the embryos collected on day 8 are still within the size range of day 7 embryos. On the other hand, it has to be considered that even when ovulation is assessed at shorter intervals in mares, there is still a considerable difference in size among equine embryos of almost similar age (e.g. Betteridge et al. J.Anat.(1982),135,1,pp.191-209; Colchen et al. J Reprod Fertil Suppl (2000) 56, 527–537). Size differences among equine embryos collected at the same age are therefore hardly avoidable. A respective sentence in this regard has been added to the discussion (lines 431-434).

It is correct that differences in mean size of the embryos stored for 24 hours were small and therefore hardly detectable in the figure. It becomes, however, apparent, that there was no clear increase in size which is in contrast to the size increase in the in vivo embryos collected on days 7 and 8. Size effects of storage at 5 and 20 degrees were assessed by non-parametric pairwise statistical comparison (Wilcoxon test). In the 5 °C group, there was an increase in size in 17 and a decrease in 3 embryos (p-value = 0.001) whereas in the 20 °C, there was in increase in size in 4, a decrease in 12 and no change in 3 embryos (p-value = 0.020). This information was added to the results section (lines 331-334).
--

Line 150: Please specify the holding medium correctly. EquiHold?

Authors’ response: We have added this information to the materials and methods. Line 153, “…embryos were kept in holding medium (EquiHold, Minitube) within an Equitainer…”
--

Results:

Line 327: “No significant difference in size among embryos collected on day 7 was detected.”

-> Can you provide the data for this? Diameter, Means, SD, p-value?

Authors’ response: We have included the information in lines 331-334.
--

Line 333: “For the determination of gene expression and DNA methylation, only blastocysts of similar development stage and ≥300 μm in diameter at collection (in total n = 52) were used.”

-> This needs to be explained in the Material and Methods section, including an update of the numbers:

2.6.      qPCR                                                  n=24, 6 per group

2.7.      gene-specific DNA methylation         n=28, 7 per group

2.8.      global methylation                            n=28, 7 per group

Please update the correct numbers, also in the Results section.     

Authors’ response: Thanks for the question. All the numbers have been checked again and are correct. We have changed the sentence to avoid misleading (line 339).
--

Figure 1: Description on the plotted parameters (mean/error/CI) and statistical tests are missing in the figure legend. For b and c the diameter axis could be shorten to visualize the significant differences. Can you add a figure for E8F versus E5C / E20C on d8? To illustrate the restricted growth of the storage embryos?

Can you mark the embryos < 300 mm in diameter that were not used for gene expression analysis and DNA methylation analysis?

Authors’ response: Thanks. We changed the figure legend as suggested. With respect to the request marking embryos < 300 um, it has to be considered that ALL embryos smaller than 300 µm were not used for gene expression and DNA methylation analysis. Therefore, there is no reason to highlight those embryos. This is clearly stated in lines 338-341.
--

Figure 2: Are these the same embryos: A/B, C,D, E/F, G/H?

Authors’ response: Yes, the embryo presented in figures A/B, C/D, E/F, G/H, respectively, is always the same before and after storage at the different temperatures. Respective information has been added to the legend.
--

Figure 3: Which values where plotted (mean / error / CI)?

Authors’ response: The legend was rewritten to allow for better understanding. Each dot represents the result from a single embryo. The plot represents the descriptive variation (mean ± S.E.M.) within the group.
--

Discussion:

Line 427: “The present results suggest that storage at 20 °C inhibits the activity of this enzyme, but in comparison to in vivo-produced embryos decreased enzyme activity is present in embryos stored at 5 °C.”

This sentence is slightly confusing… both, 5°C and 20°C embryos were significantly smaller on day 8 than in vivo derived embryos? Can you quantify the size differences between the groups (see comment for Figure 1). And better explain your suggestion for the difference between 5° and 20°?

Authors’ response: Thanks for the remark. The sentence referred to by the reviewer, now in lines 443-446, is in respect to the enzyme activity in the embryos; based on the literature, our results suggests a decrease in the enzyme activity for the stored embryos. In the present study, however, we did not directly associate the enzyme activity with embryo size. The text was slightly rewritten to avoid misunderstanding (lines 439-446).

It has been reported previously that the rapid increase in size of equine embryos starting on day 7 mainly depends on influx of fluid into the blastocoel allowed by the formation of an osmotic gradient due to activity of α1/β1 Na+/K+-ATPase (reviewed by [47]). This enzyme has been detected in horse embryos not earlier than day 8 after ovulation [48]. The present results suggest that storage at 20 °C may inhibit the activity of this enzyme, but in comparison to in vivo-produced embryos decreased enzyme activity is also present in embryos stored at 5 °C.
--

Reviewer 2 Report

This manuscript describes changes in gene expression in equine embryos stored at 5 vs 20 C compared with day 7 and day 8 embryos. The manuscript is mostly well written but there are several issues that must be addressed before being accepted for publication. 

Specific comments:

Line19: As presented, this sentence suggests only day 7 embryos were utilized in this study. Authors are requested to include that both day 7 and day 8 embryos were incubated for 24h to serve as controls for the in vitro incubations.

Lines 45-46: The authors state that temperature during storage did not affect embryo size. But in Figure 1, authors indicate that there were differences in embryo size between day 7 and day 8 stages. This should be captured in the abstract.

Lines 149-151: How were the two treatment temperatures achieved and maintained using an Equitainer? Please provide additional information here.

Lines 336-338: Figure 1 - Data presented in this figure warrants some clarification regarding the statistical analysis conducted. It appears that other than the data presented in Panel A, data presented in Panels B and C may not be statistically significant. This is especially important because a non-parametric test was used. Can the authors provide some explanation? 

Lines 357-358: Authors state that BAX expression was higher in embryos collected on day 8 compared to those collected on day 7 and/or stored. But the data present in Figure 3, contradicts this statement. In fact, BAX expression is lower in day 8 embryos.

Lines 363-364: Authors state that gene expression of POU5F1 and DNMT3a differed in E5C embryos compared to all other groups. But this is not supported by the data presented in Figure 3. For example, with respect to POU5F1, there is no difference between E7F, E5C and E20C. Likewise, there is no difference between E7C and E5C or E7C vs E5C vs E8F. Authors are requested to carefully examine all these comparisons and revise this section accordingly.

Lines 448-451: Statements made by the authors are not supported by the results presented in Figure 3.

Lines 451-453: This statement again is not supported by the data presented in Figure 3. Specifically, expression of CYP19A1, PTGES2, and DNMT1 is significantly lower in embryos stored at 20C compared to day 8 embryos.

Lines 457-460: Authors are requested to revised this sentence in the context of my comments pertaining to lines 357-358 above.

General comments: Authors are requested to revise the Discussion section after carefully examining the results section and making the necessary changes to data interpretation. 

Author Response

Response to Reviewer 2 comments.

This manuscript describes changes in gene expression in equine embryos stored at 5 vs 20 C compared with day 7 and day 8 embryos. The manuscript is mostly well written but there are several issues that must be addressed before being accepted for publication. 

Specific comments:

Line19: As presented, this sentence suggests only day 7 embryos were utilized in this study. Authors are requested to include that both day 7 and day 8 embryos were incubated for 24h to serve as controls for the in vitro incubations.

Authors’ response: Thanks for the comment. We have used only embryos collected 7 days after ovulation for storage. The embryos collected 8 days after ovulation were used as fresh control to compare the changes occurred after 24 hours of incubation with a fresh embryo. The equine embryos incubated in holding medium are capable to continue their development. Therefore, the embryos from day 8 were used as “negative controls” to determine possible changes that may occur during storage at the different temperatures evaluated in the present study.
--

Lines 45-46: The authors state that temperature during storage did not affect embryo size. But in Figure 1, authors indicate that there were differences in embryo size between day 7 and day 8 stages. This should be captured in the abstract.

Authors’ response: Thanks for the comment. Due to limit of characters, in the abstract we’ve marked the main results regarding to the different temperatures used for embryo storage (no differences were observed between 5 and 20C). The differences between fresh embryos day 7 and 8 was expected since there is a marked embryo development between these days, as reinforced in the discussion (lines 418-420; Colchen et al. J. Reprod. Fertil. Suppl. 2000, 56, 527–537; Panzani et al., Theriogenology 2014, 82, 807–814; Aurich & Budik, J. Anim. Sci. Biotechnol. 2015, 6, 1–8).
--

Lines 149-151: How were the two treatment temperatures achieved and maintained using an Equitainer? Please provide additional information here.

Authors’ response: Thanks for the question. We have rewritten the sentence as explained in line 153-156 and 175-179. The sentence now reads: For short-term storage, embryos were kept in holding medium (EquiHold, Minitube) within an Equitainer (Hamilton Biovet, Ipswich, MA, USA), with or without the freezer cans (E5C and E20C, respectively).”

The embryos kept at 5ËšC were maintained inside the Equitainer with the freezer cans provided by the company. The embryos kept at 20ËšC were kept without the freezer cans inside the Equitainer. For the experiment, the embryos at 5 and 20ËšC were not kept simultaneously in the same Equitainer. Therefore, using the data logger, we could verify the exact temperatures that were maintained inside the Equitainer for each treatment group.
--

Lines 336-338: Figure 1 - Data presented in this figure warrants some clarification regarding the statistical analysis conducted. It appears that other than the data presented in Panel A, data presented in Panels B and C may not be statistically significant. This is especially important because a non-parametric test was used. Can the authors provide some explanation?

Authors’ response: We thank the reviewer for raising this point that was also addressed by Reviewer #1. It is correct that differences in mean size of the embryos stored for 24 hours were small and therefore hardly detectable in the figure. It becomes, however, apparent, that there was no clear increase in size which is in clear contrast to the size increase in the in vivo embryos collected on days 7 and 8. Size effects of storage at 5 and 20 degrees were assessed by non-parametric pairwise statistical comparison (Wilcoxon test). In the 5 °C group, there was an increase in size in 17 and a decrease in 3 embryos (p-value = 0.001) whereas in the 20 °C, there was in increase in size in 4, a decrease in 12 and no change in 3 embryos (p-value = 0.020). This information was added to the results section to clarify this point (lines 431-434).
--

Lines 357-358: Authors state that BAX expression was higher in embryos collected on day 8 compared to those collected on day 7 and/or stored. But the data present in Figure 3, contradicts this statement. In fact, BAX expression is lower in day 8 embryos.

Authors’ response: Thanks for bringing this up. BAX expression was lower in fresh embryos collected on day 8 only when compared to stored embryos at 5C. We have corrected this sentence in the text (lines 376-377).
--

Lines 363-364: Authors state that gene expression of POU5F1 and DNMT3a differed in E5C embryos compared to all other groups. But this is not supported by the data presented in Figure 3. For example, with respect to POU5F1, there is no difference between E7F, E5C and E20C. Likewise, there is no difference between E7C and E5C or E7C vs E5C vs E8F. Authors are requested to carefully examine all these comparisons and revise this section accordingly.

Authors’ response: Thanks for bringing this up. We are truly sorry that with regard to the presentation of the results from the statistical analysis in Figure 3, some mistakes occurred. These were now corrected. The POU5F1 expression was lower in E5C embryos compared to E20C and E8F, but similar to E7F. The expression of DNMT3a of E5C differed from the other groups. We have also clarified the respective sentence in the text (lines 373-376).
--

Lines 448-451: Statements made by the authors are not supported by the results presented in Figure 3.

Authors’ response: Please see above. The results with regard to data presented in Figure 3 were corrected as requested by this reviewer in the previous question. The respective text in the discussion is no longer in any contradiction to the results.
--

Lines 451-453: This statement again is not supported by the data presented in Figure 3. Specifically, expression of CYP19A1, PTGES2, and DNMT1 is significantly lower in embryos stored at 20C compared to day 8 embryos.

Authors’ response: Please see above.
--

Lines 457-460: Authors are requested to revised this sentence in the context of my comments pertaining to lines 357-358 above.

Authors’ response: Please see above.
--

General comments: Authors are requested to revise the Discussion section after carefully examining the results section and making the necessary changes to data interpretation. 

Authors’ response: We thank the reviewer for checking these points with such great care. In figure 3, the presentation of results from the statistical analysis was carefully checked and corrected. The text in the discussion is now in accordance with the results.
--

Round 2

Reviewer 2 Report

The authors have adequately addressed my comments.